# Antibacterial Activity of Bis(4-aminopyridinium) Compounds for Their Potential Use as Disinfectants

**DOI:** 10.3390/molecules30193962

**Published:** 2025-10-02

**Authors:** Carolina Arriaza-Echanes, Claudio A. Terraza, Mateus Frazao, Sebastián Reyes-Cerpa, Loreto Sanhueza, Pablo A. Ortiz

**Affiliations:** 1Centro de Nanotecnología Aplicada, Facultad de Ciencias, Ingeniería y Tecnología, Universidad Mayor, Camino La Pirámide 5750, Huechuraba 8580745, Chile; carolina.arriaza@mayor.cl; 2Research Laboratory for Organic Polymers (RLOP), Department of Organic Chemistry, Pontificia Universidad Católica de Chile, Santiago 7820436, Chile; cterraza@uc.cl; 3Centro de Genómica y Bioinformática, Facultad de Ciencias, Ingeniería y Tecnología, Universidad Mayor, Santiago 8580745, Chile; mateus.frazao@mayor.cl (M.F.); sebastian.reyes@umayor.cl (S.R.-C.); 4Escuela de Biotecnología, Facultad de Ciencias, Ingeniería y Tecnología, Universidad Mayor, Santiago 8580745, Chile; 5Núcleo de Química y Bioquímica, Facultad de Ciencias, Ingeniería y Tecnología, Universidad Mayor, Camino La Pirámide 5750, Huechuraba 8580745, Chile; 6Escuela de Ingeniería en Medio Ambiente y Sustentabilidad, Facultad de Ciencias, Ingeniería y Tecnología, Universidad Mayor, Camino La Pirámide 5750, Huechuraba 8580745, Chile

**Keywords:** organic salt, mechanism of action, multidrug-resistant bacteria, bis(4-aminopyridinium)alkanes

## Abstract

The following study presents the initial evaluation (solubility, thermal stability, antibacterial activity, and cytotoxicity) of a series of previously described organic salts, derived from the bis(4-aminopyridinium) cation with different chain lengths, for their potential use as hospital disinfectants. Of the salts studied, those with chain lengths between 2 and 10 carbon atoms (C_2_–C_10_) showed high solubility in water, methanol, and DMSO. All salts exhibited high thermal stability, showing a thermal decomposition temperature (T_5%_) above 330 °C. Antibiotic susceptibility testing of the studied *E. coli*, *S. aureus*, and *S. typhimurium* strains confirmed their resistance to different classes of commonly used clinical antibiotics, validating their selection. During the determination of antibacterial activity, the long-chain salts (C_10_ and C_12_) showed the greatest activity, with minimum inhibitory concentrations (MICs) from 31.2 μg/mL to 62.5 μg/mL in all the strains studied. Given the high activity of C_10_ and C_12_, their cytotoxicity was assessed in HeLa cells. They exhibited no cytotoxic effects after 12 h and only about 5% cytotoxicity after 24 h. Furthermore, the cell viability assay of the most active and water-soluble salt, C_10_, showed that this salt can interact with the bacterial cytoplasmic membrane, increasing its permeability in both Gram-positive and Gram-negative bacteria. However, these results cannot rule out the possibility that this salt may have more than one site of action within the bacterial cell.

## 1. Introduction

Healthcare-associated infections (HAIs) are infections that affect a patient during their admission to a hospital or healthcare facility, which were not present at the time of admission and may manifest after discharge [1]. HAIs are considered one of the biggest health problems worldwide [2]. In high-income countries, they affect 7 out of every 100 patients admitted to a healthcare facility, and in low- or middle-income countries, 15 out of every 100 may present HAIs during their hospitalization [3]. These infections cause the death of 700,000 people per year worldwide, and it is estimated that, if no measures are taken, by 2050, this figure will increase to 10 million [4]. In this context, inert surfaces are an important link in the transmission chain of microorganisms such as bacteria. According to the World Health Organization, implementing environmental cleaning and disinfection measures can reduce the incidence of hospital-acquired infections by up to 70%, and for this reason [3], disinfectants play a fundamental role.

Disinfectants are defined as a product that reduces the number of microorganisms in or on a matrix through their irreversible action to a level considered adequate for a defined purpose (European Chemicals Agency: ECHA). These substances are generally active against a broad spectrum of Gram-positive and Gram-negative bacteria and are nonspecific against various microorganisms [5]. Among the most used disinfectants are alcohols, such as ethanol and isopropanol, which are characterized by their rapid action, residue-free activity, and flammability, as well as decreased effectiveness in the presence of organic matter [6,7]. Aldehydes, such as glutaraldehyde and orthophthaldehyde, are highly effective; however, they have toxic and skin-irritating effects [6]. Oxidizing agents such as peroxides (hydrogen peroxide and peracetic acid) have been used for their broad antibacterial spectrum and low environmental impact, although they are corrosive and expensive [6,8]. Phenols are economically viable and useful on inanimate surfaces, but cannot be used on human tissue due to their toxicity. Halogens, such as iodine and chlorine, are very effective and inexpensive, but generate toxic vapors, are corrosive, and their effectiveness decreases over time [7,8]. Finally, quaternary ammonium compounds (QACs) are used for their easy application, non-corrosiveness, and effectiveness against bacteria, viruses, and fungi [7,9]. This last group is of special interest since its chemical structure is composed of a quaternary ammonium group, an active group that interacts with the cytoplasmic membrane, while its alkyl chain, with 12 to 16 carbon atoms, improves antibacterial activity [10]. Despite this, its efficacy has been questioned due to non-standardized evaluation methods, massive use, and its prolonged permanence in the environment, which induces bacterial resistance. This fact compromises its effectiveness and raises concern about the possible impact they may have on health and the environment [7,11,12,13]. Additionally, the excessive use of QACs is expected to generate cross-resistance with antibiotics, since disinfectants and antibiotics are evaded by bacteria by common mechanisms [13]. Although there is a wide variety of disinfectants, none of these can be used universally, since each one has some limitations, such as low effectiveness, toxicity, corrosiveness, irritability, or high cost. That said, new molecules are required for the development of disinfectants that can contribute to the prevention and mitigation of HAIs [8]. In general, an ideal disinfectant is expected to have high water solubility, a broad spectrum of activity, and little or no toxicity. Furthermore, it must be stable throughout its shelf life, not react with or be inactivated by organic matter, have rapid action, penetration capacity, be non-corrosive, be readily available, have a good cost-risk-benefit ratio, and be environmentally friendly [7].

Today, the monitoring and control of HAIs are crucial for clinical safety and a priority within quality and patient safety policies [11]. HAIs have been described in association with bacteria of the genera *Staphylococcus*, *Enterococcus*, *Clostridium*, and *Enterobacteriaceae*, among others [4]. Therefore, the study and development of more effective, safe, and broad-spectrum disinfectants against antibiotic-resistant bacteria that hinder medical treatment is of vital importance to combat and control HAIs.

Organic salts, such as ionic liquids (ILs), have emerged as a promising alternative for the development of new disinfectant agents. These compounds have been shown to possess diverse biological activities, including anticancer, antifungal, and antibacterial [14,15,16,17].

ILs are organic salts typically composed of a bulky organic cation and an inorganic or complex anion. Unlike other organic salts, these ILs have melting points equal to or lower than 100 °C.

The presence of hydrophobic fragments combined with ionic charges confers many ILs with surfactant properties, further expanding their applications in biomedicine and biotechnology [18]. In terms of antibacterial activity, several ILs have been shown to be active against clinically relevant pathogens such as *S. aureus*, *E. coli*, *K. pneumoniae*, *Enterococcus faecium*, *Micrococcus luteus*, *Staphylococcus epidermidis*, and *Streptococcus mutans*, without presenting significant toxicity in mammalian cells at therapeutic concentrations [19]. However, the biological activity of an IL can vary considerably depending on its chemical structure, highlighting the importance of understanding the relationship between molecular structure and antibacterial activity. Key structural factors influencing the antibacterial activity of ILs include the length of the alkyl chain, the nature and aromaticity of the cation, the presence of specific functional groups, and the nature of the anion [20]. Cationic compounds, such as phosphonium, ammonium, imidazolium, and pyridinium, have demonstrated antibacterial activity against a broad spectrum of bacteria, leading to their use as disinfectants and cleaning agents in the food, pharmaceutical, and healthcare industries [19]. Examples of this are organic salts and ILs containing pyridinium cations in their structure. These compounds have been shown to possess bactericidal activity, positioning them as a viable alternative for the development of new disinfectants [21]. Furthermore, studies conducted by the Aggarwal [22] and Govindaraj [23] groups on a series of mono- and di-cationic pyridinium compounds showed that these compounds possess low cytotoxicity, making them viable for biomedical applications.

In general, the mechanism of action of these compounds is based on their interaction with the cell membrane. Being amphiphilic, they insert into the lipid bilayer, causing disorder, altering permeability, and ultimately destroying the cell [15,24,25,26]. Since this is a physical effect, it considerably hinders bacterial adaptation, reducing the possibility of developing antibacterial resistance. Furthermore, structural modifications of these types of cationic compounds, such as the length of the alkyl chain, can significantly modify their antibacterial activity [27].

In addition to their antibacterial properties, these compounds exhibit unique physicochemical characteristics, such as their low or zero vapor pressure, which means they are neither volatile nor flammable [28,29], a substantial advantage when compared to halogens or alcohols. Their high thermal and chemical stability allows for safe storage and handling [30,31,32], along with the ability to recover and be reused relatively easily [33]. Their low corrosiveness makes them compatible with various hospital-grade materials, such as stainless steel [34,35]. Their multiple solvation layers allow them to interact efficiently with different types of molecules [16].

On the other hand, like drugs, the effectiveness of a disinfectant depends largely on its bioavailability, which is directly related to its solubility and permeability. In the case of antibacterial ILs, low water solubility can significantly limit their bactericidal activity, since it makes it difficult to reach effective concentrations at their site of action [36,37]. Therefore, improving the solubility and dissolution rate of these compounds is one of the main challenges for their development [38]. An example of these improvements is the work carried out by Israr et al. [32], who synthesized pyridinium-based dicationics with different hydrophilic and hydrophobic spacer groups. Their results show that these compounds have remarkable solubility in various aqueous environments (seawater, deionized water, and formation water) at room temperature, with no phase separation or turbidity observed after three weeks. Taking this into account, organic salts such as ILs and salts with pyridinium cations in combination with variable alkyl chains emerge as a promising alternative due to their unique physicochemical properties and high capacity to interact with bacterial membranes [39,40].

Based on the above, the objective of this study was to evaluate the relationship between the molecular structure, solubility, and antibacterial activity of a series of organic salts derived from 4-aminopyridine, against strains of *S. aureus*, *S. aureus* MRSA, *E. coli* and *Salmonella enterica serovar Typhimurium* for their application as hospital disinfectant, and to evaluate the cytotoxicity of the most active compounds.

## 2. Results and Discussion

### 2.1. Synthesis and Spectroscopy Characterization of Organic Salts

The synthesis of the selected salts (Figure 1) was carried out similarly to that described by Bunting [41], Liu [42], Patil [43], Watanabe [44], Walker [45], Ng [46], and Abeywickrama [47], between 1861 and 2018. However, unlike what was previously reported [41,42,43,44,45,46,47], the reactions were carried out in DMF to keep the 4-aminopyridine in solution and thus avoid its presence in the crude and purified product. As in these studies, the syntheses were carried out by a Menshutkin reaction, where the nucleophilic capacity of 4-aminopyridine was favored by the electron-donating effect of the amino group located in the para position, as described by Walker [45]. Because of this, during the first 30 min of stirring at room temperature, the formation of a white or yellow solid was observed. Additionally, upon initial heating, depending on the solubility, an increase in the amount of precipitate was observed, which increased with decreasing reaction temperature.

In this case, the molecular structure of the salts was corroborated by FT-IR, ^1^H, ^13^C, and DEPT 135° NMR spectroscopy. In the FT-IR spectra, the most representative bands of the common bonds were N-H (3223–3493 cm^−1^), aromatic C-H (3043–3192 cm^−1^), aliphatic C-H (2856–2992 cm^−1^), C=N (1640–1666 cm^−1^), C=C (1502–1634 cm^−1^), C-N (1171–1204 cm^−1^), *para*-disubst. (827–877 cm^−1^), and Py ring (610–669 cm^−1^). While the ^1^H NMR spectra showed the presence of hydrogens belonging to the pyridine ring (3 and 4) and a band centered between 4.12 and 5.36 ppm, which was assigned to the hydrogens of the methylene groups called (5). Generally, these nuclei are observed at higher fields, but due to the inductive attraction effect of the pyridinium group formed during synthesis, they move to a lower field. Another important band was the one associated with the hydrogens of the amino groups (1). However, these nuclei were only evident in the spectra of those salts analyzed in DMSO-*d_6_* (C_8_, C_10_, and C_12_), the solvent used, given the lower solubility of these compounds in deuterated water.

In these cases, a singlet was observed that integrated for four between 8.14 and 8.19 ppm. This is due to the higher acidity of D_2_O compared to DMSO-*d_6_*, which facilitates the exchange of hydrogens from the salts for deuterium nuclei in the solvent.

The detailed spectroscopic characterization of each salt is presented later in Materials and Methods, and the spectra obtained during the analyses are shown in Appendix A of the Supporting Information.

### 2.2. Solubility

The results of the solubility tests performed on the synthesized organic salts are shown in Table 1. All the salts that present aliphatic groups (flexible) in the central part of their structure showed high solubility in water. This fact was of great importance, as it allows for the preparation of aqueous solutions; avoids the use of solvents that could be toxic or interfere with the microbiological analysis of the molecules; and, above all, favors their biological activity. It was also possible to observe how most salts were soluble in highly polar solvents, which can make hydrogen bonds or have a high ionic character, such as MeOH, EtOH, DMF, and especially DMSO. This characteristic is derived from the presence of pyridinium ionic groups. It should be noted that, although this behaviour allows estimating the solvent to be used for spectroscopic characterization, it is not conclusive. Since at the time of preparing the solutions to carry out the analyses, the longer-chain salts (C_8_ to C_12_) were insoluble in deuterium oxide but soluble in deuterated DMSO. The latter indicates that if organic salts of similar structure and a longer chain are used, their solubility in water will decrease, as is the case for C_12_ with respect to C_10_. C_12_ was only soluble in deuterium oxide at a temperature above 40 °C and could recrystallize when the temperature decreased.

As expected, and given the aromatic contribution of the central portion of C_Ph_, this salt presented the lowest solubility in the solvents tested, reaching only partial solubility in DMSO and water when the mixture was exposed to a temperature close to 40 °C. All the samples were insoluble, even when heated, and remained insoluble even with the application of solvents of lower polarity or those unable to form hydrogen bonds with the salt.

### 2.3. Thermal Properties

The TGA and DTGA thermograms obtained for each salt are shown in Appendix A of the supporting information, while the main results of the thermal analyses are summarized in Table 2. These analyses were carried out to determine the melting temperature of each salt and its thermal behaviour, which provides information for the drying and storage processes of these salts.

All the salts begin their decomposition process at temperatures above 300 °C (T_onset_) with a tendency to decrease this parameter as the aliphatic chain grows. In general, this high thermal resistance was associated with the pyridinium groups and the ionic interactions present in each molecule.

The mass loss values associated with 5%, 10% and 50% were centered in the ranges 331–400 °C, 338–404 °C, and 358–416 °C, respectively, showing the same tendency to decrease as the amount of methylene groups increases. The latter was consistent with the relationship between the aliphatic and aromatic segments in the salts, since, as the methylene groups increase, the contribution of the aliphatic segment to their thermal properties increases, a section that requires less energy to degrade its links.

The DTGA curves (Appendix A) show that the degradation process of these salts occurred in two stages, with maximum degradation rate values that can be averaged by discarding C_2_ from the series, a sample that presents significantly higher values. The first stage, near 370 °C, corresponds to the degradation of the aliphatic sections of each salt, while the second stage, centered at ≈510 °C, degrades the aromatic moieties. From these results, an optimal drying temperature of 200 °C was established. A temperature that allowed for the removal of occluded water and DMF was used both in the synthetic process and in the purification by recrystallization of some of them. Additionally, this enables the storage of the salts at room temperature, without risk of degradation or the need for additional refrigeration systems. Finally, based on this information, it was possible to apply the temperature program that allowed us to obtain the melting temperature of the salts, free from thermal phenomena derived from the isolation and purification processes.

Before beginning the DSC analysis of the salts, it was expected that there would be a direct relationship between the length of the central hydrocarbon segment and the melting temperature. This trend should show a progressive decrease in temperature due to the lower contribution of the ionic group and the higher contribution of the hydrocarbon chain. While this was observed for C_2_, C_3_, and C_4_, C_5_ did not show a melting point, and although a decrease in melting temperature was again observed from C_6_ to C_12_, from C_6_ onwards, the values were higher than those obtained for C_3_. This suggests that the length of the aliphatic hydrocarbon moiety was a factor that determined the melting temperature. A second element to be analyzed also has a strong influence on this parameter and is related to the flexibility of the central chain and, therefore, to its conformation and/or arrangement. Although what was observed may be of interest to be analyzed and investigated, for the moment, this is outside the context of what is reported in this work.

Contrary to expectations, the incorporation of the xylenyl moiety into C_Ph_ (-CH_2_PhCH_2_-) resulted in a lower melting point compared to C_2_. While the phenyl ring, the only structural difference between the two salts, can provide π–π stacking interactions, it can also cause difficulties in the packing of the structures, resulting in a lower melting point than C_2_ but higher than the rest of the salts.

### 2.4. Antibiotic Susceptibility

Antibiotic susceptibility tests were performed to detect bacterial resistance based on their phenotype. This allowed for the selection of strains resistant to different classes of antibiotics. The strains selected in this step were used to determine the antibacterial activity of the salts.

The assays performed are those recommended by the Clinical & Laboratory Standards Institute (CLSI) for this purpose. It is important to note that these tests made it possible to classify the strains as susceptible or resistant, but they cannot determine the mechanisms involved in the observed resistance.

The results of the susceptibility tests performed on the strains in our strain collection, *S. aureus* ATCC 6538, *S. aureus* MRSA 97-7, *E. coli* ATCC 25922, and *S. Typhimurium* ATCC 14028s strains, are shown in Table 3. The strain of *S. aureus* ATCC 6538 was susceptible to all the antibiotics tested except penicillin. By contrast, *S. aureus* MRSA 97-7, *E. coli* ATCC 25922, and *S. Typhimurium* ATCC 14028s were resistant to more than three classes of antibiotics according to the criteria established by CLSI. This indicates that the strains of *S. aureus* MRSA 97-7, *E. coli* ATCC 25922, and *S. Typhimurium* ATCC 14028s used should be classified as multi-resistant bacteria [48].

All these strains were resistant to betalactamic antibiotics, cephalosporins, lincomycin, tetracyclines, and macrolides.

The site of action of betalactamic and cephalosporin antibiotics is the bacterial cell wall. They act by inhibiting the final stages of peptidoglycan (cell wall) synthesis. Meanwhile, tetracycline enters the bacterial cytoplasm through an energy-dependent process by reversibly binding to the 30S subunit of the ribosome, blocking the access of aminoacyl-tRNA complexes and preventing protein synthesis [49]. On the other hand, macrolides inhibit protein synthesis by binding to the 50S subunit of the ribosome [50]. *S. aureus* MRSA 97-7 strain also showed resistance to ciprofloxacin, an antibiotic belonging to the fluoroquinolone class. The site of action of fluoroquinolones is DNA, which acts on enzymes: gyrase and isomerase type IV, forming a ternary complex between the fluoroquinolone, the enzyme, and DNA, thus inhibiting the replication process [49,50].

In the case of the *E. coli* ATCC 25922 strain, it is generally considered susceptible to most antibiotics and is used in numerous studies as a control strain to evaluate the antibiotic susceptibility of clinical isolates of *E. coli*. However, this strain can develop resistance to these drugs through different processes: mutations by physical agents (e.g., UV radiation), transfer of resistance genes (conjugation, transformation, and transduction), or through selective pressure from antibiotics (acquisition citations) [51,52]. An example of the variability in the resistance profile of the *E. coli* ATCC 25922 strain is the work carried out by Mursvida et al. [53]. They studied its antibiotic susceptibility using the disk diffusion method, determining that it presented resistance to different antibiotics, including cefotaxime, cefixime, ceftriaxone, and chloramphenicol.

### 2.5. Evaluation of the Antibacterial Activity of the Organic Salts

The antibacterial activity against *S. aureus* ATCC 6538, *S. aureus* MRSA 97-7, *E. coli* ATCC 25922, and *S. Typhimurium* 14028s strains of a set of organic salts varying in alkyl chain length over a range of 2 to 12 carbon atoms, including C_Ph_, was evaluated. As shown in Table 4, the salts C_Ph_, C_2_, C_3_ and C_5_ did not present antibacterial activity at the maximum concentration studied (2000 µg/mL) in all the strains tested, except for *S. aureus* ATCC 6538 (strain susceptible to all the antibiotics tested), which showed a MIC of 1000 µg/mL for the salts C_Ph_ and C_2_, which does not represent a significant antibacterial effect. On the other hand, the results obtained for C_4_ showed relatively low MIC values, but well above those observed for C_10_ and C_12_. This result could be due to variations in the molecule’s conformation, derived from the length of the spacer chain, which would cause alterations in the charge distribution, making its behavior similar to that of C_6_ and C_8_. However, although this was an interesting result, its analysis will not be further explored because it is beyond the scope of this work.

In general, starting from compound C_6_, antibacterial activity increases as the alkyl chain length increases in all strains tested, with the highest activity being recorded for compounds C_10_ and C_12_. The MICs determined for these compounds were 31.2 µg/mL for *S. aureus* ATCC 6538, *S. aureus* MRSA 97-7, and *S. Typhimurium* 14028s, while for *E. coli* ATCC 25922, the MICs increased slightly at a concentration of 62.5 µg/mL. The results obtained show that the compounds C_10_ and C_12_ have significant antibacterial activity, both for Gram-positive bacteria (*S. aureus* ATCC 6538, *S. aureus* MRSA 97-7) and Gram-negative bacteria (*S. Typhimurium* 14028s, *E. coli* ATCC 25922) studied.

The antibacterial activity of ILs has been widely described [18,19,25]. An example is the work done by Garcia et al. [54] who determined the antibacterial activity of new cholinium-based ionic liquids against *S. aureus* MRSA strains. The MICs varied in a range of 16–32 µg/mL. These MICs were very similar to those obtained with compounds C_10_ and C_12_ against the strain *S. aureus* MRSA 97–7. Pałkowski et al., 2022 [55], describe the antibacterial activity of new imidazolium-based ionic liquids, determining that compounds containing 10 to 12 carbon atoms in the alkyl chain exhibit the highest antibacterial activity, but this activity decreases as the chain length increases above 12 carbon atoms.

In addition, Kuhn et al. [56] reported that the antibacterial activity of ILs increases with the length of the alkyl chain. Chains of 12–14 carbon atoms showed greater activity than ILs with chains of more than 16 or less than 10 carbon atoms. All these results coincide with those obtained in this study, where organic salts with 10 and 12 carbon atoms were those that presented greater antibacterial activity for both Gram-positive and Gram-negative bacteria (MICs 31.2–62.5 µg/mL).

Furthermore, it has been described that this type of organic salt is more active against Gram-positive bacteria than against Gram-negative bacteria. Xu et al. [57] synthesized organic salts based on hydrophobic esters, determining that those compounds with alkyl chains of 16–18 carbon atoms presented greater antibacterial activity against *S. aureus* (MIC < 2 µg/mL), while less hydrophobic compounds with alkyl chains of 10–14 carbon atoms were more active against *E. coli* (MIC = 32 µg/mL). This is because Gram-positive bacteria have only one phospholipid membrane and a thick peptidoglycan wall, while Gram-negative bacteria have a more complex structure composed of two phospholipid membranes, separated by a thin layer of peptidoglycan. The outer membrane is composed of proteins and lipopolysaccharides (LPSs), which make it difficult for antibacterials to enter [18,58,59,60]. Furthermore, it has been described that this type of compound favors the disruption of the cell wall, which would also explain its effect on Gram-positive bacteria [16,61].

However, in this study, no differences were observed between Gram-positive and Gram-negative bacteria when using compounds with different alkyl chain lengths.

Although the antibacterial activity of ILs depends on the chain length, it is also on other factors such as charges and the chemical composition of the anions and cations [50]. The most accepted mechanism of action of this type of compound is related to the electrostatic attraction of opposite charges and hydrophobic interactions between negatively charged cell membranes and positively charged nitrogen atoms. In these cases, the cation interacts strongly with the polar head of the phospholipid, remaining close to the water-membrane interface, while its alkyl side faces the lipid zone of the membrane where long alkyl chains tend to insert more deeply [16].

In the initial phase, the cationic molecules are attracted by the phospholipids, introducing a net positive charge, depolarizing the bacterial cytoplasmic membrane, which produces distortions in it [62]. The divalent magnesium and calcium cations that stabilize the membrane surface are replaced by cationic IL molecules, which decreases membrane fluidity. The alkyl chains present in the molecule penetrate the inner part of the lipid bilayer, inserting and integrating into the membrane structure [62]. This process leads to the formation of transient pores or channels that increase permeability and alter their physical properties [24,62]. Perforation progressively causes leakage of metabolites from the intracellular space, the release of potassium ions along with other cytoplasmic constituents, and the alteration of intracellular biochemical processes [48,49,50]. As a result, the membrane completely loses its physiological barrier and its osmoregulatory functions, proteins and nucleic acids are degraded, while the autolysis pathway is activated, ultimately leading to cell death [55].

Other mechanisms of action described for this type of compound include enzyme inhibition [58,59,60], induction of lipid peroxidation, and the induction of oxidative stress [63].

### 2.6. Live/Dead Assays

This assay was conducted to determine whether one of the sites of action of these compounds was the cytoplasmic membrane, given that it is often a common target for these types of compounds. For this purpose, the compound with the highest antibacterial activity against both Gram-positive and Gram-negative bacteria and the highest solubility in water (C_10_) was selected. The compound was tested at concentrations of ½ the MIC and ¼ the MIC [16,24,38,60]

For these tests, the strain *S. aureus* MRSA 97-7 was selected as a representative of Gram-positive bacteria, and *E. coli* ATCC 25922 was used as a Gram-negative model. For the determination of membrane integrity in both strains, the BacLight Kit [64] was used. This contains the fluorescent dyes SYTO 9 and propidium iodide (PI), which can be used to distinguish live and dead cells based on their membrane integrity. Both dyes bind nucleic acid, resulting in an increased fluorescent signal. SYTO 9 is membrane-permeable and can enter all cells, resulting in green fluorescence. PI is membrane-impermeable and can only enter cells with compromised membranes (regarded as dead), resulting in red fluorescence.

Figure 2 shows the effect of compound C_10_ on the permeabilization of the cytoplasmic membrane. In the case of *E. coli* (Figure 2a), the proportion of dead cells increased as the salt concentration increased. When cells were treated with compound C_10_ at a concentration of ¼ of the MIC (15.6 µg/mL), the green/red fluorescence ratio had a value of 3.57, which is lower than the ratio observed in untreated cells (C-100% live cells), with a value of 4.59. If the C_10_ concentration increased to ½ of the MIC value (31.3 µg/mL), the green/red fluorescence ratio decreased to 1.02, a value like 0.84 obtained from the positive control (C+, 100% dead cells). A similar trend was observed when *S. aureus* MRSA 97-7 was treated, but the effect of compound C_10_ on this strain was greater. The decrease in the green/red fluorescence ratio at concentrations of ¼ of the MIC (7.8 µg/mL) and ½ of the MIC (15.6 µg/mL) was 2.7 and 1.9, respectively. These values turned out to be close to those obtained with the positive control, which showed a green/red fluorescence ratio value of 0.17. Therefore, it is suggested that one of the sites of action of the salts studied was the bacterial cytoplasmic membrane, increasing its permeability in both Gram-positive and Gram-negative bacteria.

In the literature, the activity observed in these compounds is mainly due to their permanent positive charge, which facilitates the formation of electrostatic interactions with the negatively charged bacterial membrane. This causes destabilization and permeabilization of the lipid bilayer, progressively triggering the leakage of metabolites from the intracellular space, releasing potassium ions along with other cytoplasmic components, and altering intracellular biochemical processes [24,38,60,61,62]. As a result, the membrane completely loses its physiological barrier and osmoregulatory functions; proteins and nucleic acids are degraded, while the autolysis pathway is activated, ultimately leading to cell death [23,25,63]. Based on this, it is suggested that these compounds interact with the cytoplasmic membrane similarly, increasing its permeability in both Gram-positive and Gram-negative bacteria. However, further molecular analysis is required to confirm this.

In this regard, it has been described that imidazolium, ammonium, piperidinium, and benzimidazolium groups are commonly used for the synthesis of these salts. Studying the interaction of these groups with cytoplasmic membranes has shown that when the charges are not protected, they cause a reorientation of the lipid fraction of the membrane and increase its insertion into it. This effect is even more pronounced in the case of large cations such as benzimidazolium and piperidinium [16].

Although it was determined that one of the sites of action for this salt is the cytoplasmic membrane, it cannot be ruled out that these salts have another cellular target in the strains under study and that they contribute to the observed antibacterial activity. However, since the antibacterial activity of these compounds relies on their interaction with the membrane, a less selective mechanism, the development of bacterial resistance is less likely. Furthermore, the ability to fine-tune the molecular structure of these compounds to optimize their antimicrobial activity increases their potential for the development of new antimicrobial disinfectants.

### 2.7. Evaluation of the Cytotoxicity of C_10_ and C_12_ Salts in HeLa Cells

The cytotoxic effect of C_10_ and C_12_ was evaluated for 6, 12, and 24 h at different concentrations in HeLa cells, using the LDH assay as an indicator of cell death (Figure 3). The results showed that neither C_10_ nor C_12_ induced a cytotoxic effect in HeLa cells after 6 and 12 h of exposure to these salts at all concentrations tested (62.5–7.8 µg/mL), and no significant differences were observed between these concentrations. However, when HeLa cells were exposed to the highest concentrations of C_10_ and C_12_ (62.5 and 31.2 µg/mL) for 24 h, both salts showed a cytotoxic effect that did not exceed 5%.

Govindaraj et al. [23] studied the effect of a series of monocationic and dicationic pyridinium bromides on fibroblast cell lines. They determined that their compounds are not cytotoxic in the concentration range used in their study (10–80 µg/mL), a range similar to the MICs of the organic salts analyzed in this work. Therefore, the toxic effect should depend on the concentration and the safety level associated with the application.

The cytotoxicity results in HeLa cells at 6 and 12 h of exposure, and performed with the MICs (62.5 and 31.2 µg/mL) obtained in this study for compounds C_10_ and C_12,_ showed that these compounds are not cytotoxic. This indicates that both C_10_ and C_12_ are good candidates for the development of a hospital disinfectant, since they present significant antibacterial activity and little or no toxicity towards these human cells.

Among the general requirements for submitting efficacy tests for biocides, disinfectants, or sanitizers, the following stand out: products with antimicrobial action must be evaluated at the lowest dilution with proven efficacy [7]. In the case of this study, C_10_ and C_12_ were not cytotoxic at MIC concentrations at 6 and 12 h of exposure, while at 24 h, only 5% of exposed cells died.

Another important requirement for the development of this type of formulation is the exposure period. For a product with antimicrobial action, the time must be as short as possible. Another important requirement for the development of this type of product is the exposure time. For a product with antimicrobial action, this time must be as short as possible. According to the Association of Official Analytical Chemists (AOAC) and the Centers for Disease Control and Prevention (CDC), the maximum exposure time should not exceed 10 min (AOAC Dilution Method for the Analysis of Disinfectants and Guideline for Disinfection and Sterilization in Healthcare Facilities) [7]. Therefore, and in accordance with this point, the studied salts could potentially be used for these purposes.

Furthermore, the characteristics of an ideal disinfectant include high water solubility, a broad spectrum of activity, low or no toxicity to humans, stability, rapid action, penetration capacity, and residual action, among others. Of these properties, C_10_ has high water solubility, a broad spectrum of antibacterial activity, and low or no toxicity. This last point considers that a disinfectant must have an exposure time of no more than 10 min. Therefore, C_10_ is a good candidate for further testing for the development of a disinfectant, which would include: evaluation of antibacterial activity against strains such as *Pseudomonas aeruginosa* and *K. pneumoniae*, along with analyses to determine the effect of pH on its chemical, physical, and biological properties; compatibility with materials (corrosion tests), surfactants, and organic matter; and degradation testing. All of this is based on methodologies validated by the AOAC [7] and the Environmental Protection Agency (EPA) (Germicidal Aerosol Products as Disinfectants [GSPT]: *Staphylococcus aureus*, *Pseudomonas aeruginosa*, and *Salmonella enterica* Tests) [65].

## 3. Materials and Methods

### 3.1. Materials and Equipment

4-Aminopyridine (4-APy), 1,2-dibromoetane, 1,3-dibromopropane, 1,4-dibromobutane, 1,5-dibromopentane, 1,6-dibromohexane, 1,8-dibromooctane, 1,10-dibromodecane, α,α’-dibromo-*p*-xylene, *N,N’*-dimethylformamide (DMF), methanol (MeOH), ethanol (EtOH), dimethylsulfoxide (DMSO), acetonitrile (AcCN), acetone, tetrahydrofuran (THF), chloroform and diethyl ether (Et_2_O) were purchased from Sigma Aldrich-Merck (Milwaukee, WI, USA). All chemicals were used as received.

FT-IR-ATR (ZnSe) spectra were recorded on a Spectrum Two (Perkin Elmer, Hopkinton, MA, USA) spectrophotometer over the range of 4000–600 cm^−1^. ^1^H, ^13^C, and DEPT 135° NMR spectra were recorded on a Bruker Advance III-400 Hz spectrometer (Bruker Corporation, Karlsruhe, Germany) using deuterated dimethylsulfoxide (DMSO-*d_6_*) and deuterium oxide (D_2_O) as solvent. The thermal stability of organic salts was evaluated using a TGA-50 SHIMADZU thermogravimetric analyzer (Columbia, SC, USA). Analysis was performed using a temperature range of 30–800 °C, under a nitrogen atmosphere and a heating ramp of 20 °C/min. The melting point (Mp) of the organic salts was determined using a PerkinElmer DSC 4000 differential scanning calorimeter from the second scan (PerkinElmer, Hopkinton, MA, USA). The measurements were from 15 °C to 415 °C at a speed of 20 °C/min, with a nitrogen atmosphere with a flow of 20 mL/min.

### 3.2. Organic Salts Synthesis

60 mmol of 4-APy was placed in a 250 mL flask. Subsequently, 90 mL of DMF was added, and the mixture was stirred until 4-APy was completely dissolved. To the solution formed, 30 mmol of the respective dibromide was added, leaving the new solution stirring for 30 min. After this time, the reaction was heated and stirred for 24 h at 100 °C. After the reaction time, the mixture was cooled to room temperature, and the solvent was removed by distillation under reduced pressure. The resulting solid was dried at 100 °C under reduced pressure for 24 h and then recrystallized in a solvent or mixture of solvents selected for each substance (MeOH, EtOH, DMF, or MeOH-H_2_O mixture).

**1,1′-(Ethane-1,2-diyl)bis(4-aminopyridin-1-ium) bromide [C_2_(4-NH_2_Py)_2_][Br]_2_ (C_2_)** [41,42]



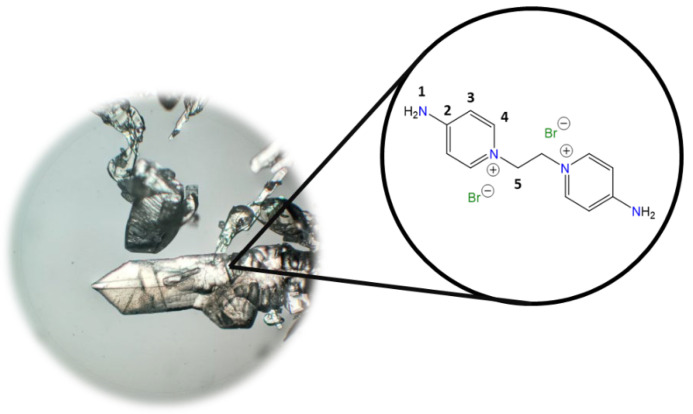



Chemical structure (right) and the optical micrograph (left) of the compound obtained for C_2_.

Yield: 91%. Solid pale yellow. Mp: 397 °C. FT-IR-ATR (ZnSe, ν, cm^−1^): 3328, 3258 (N-H); 3170, 3146, 3061 (C-H, arom.); 2982, 2092 (C-H, aliph.); 1662 (C=N); 1634, 1561, 1544, 1511 (C=C); 1204 (C-N); 842 (*p*-subst); 624 (Py-ring). ^1^H NMR (D_2_O, δ, ppm): 7.86 (d, *J* = 7.2 Hz, 4H, **4**); 6.87 (d, *J* = 7.1 Hz, 4H, **3**); 4.66 (s, 4H, **5**). ^13^C NMR (D_2_O, δ, ppm): 159.20 (**2**); 142.37 (**4**); 110.36 (**3**); 57.13 (**5**). (Spectra in Appendix A).

**1,1′-(Propane-1,3-diyl)bis(4-aminopyridin-1-ium) bromide [C_3_(4-NH_2_Py)_2_][Br]_2_ (C_3_)** [43,44]



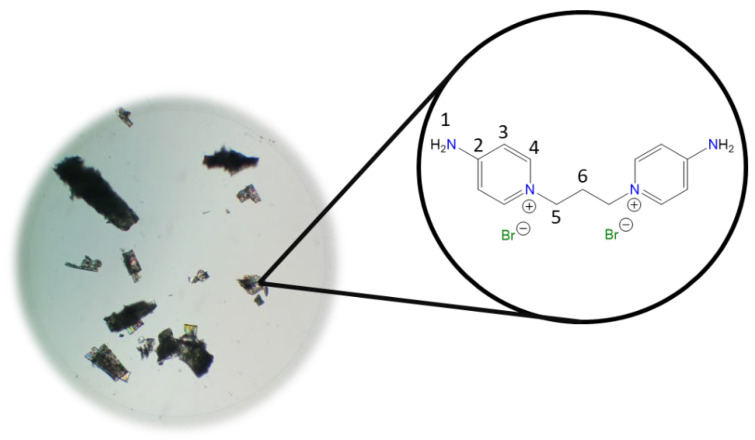



Chemical structure (right) and the optical micrograph (left) of the compound obtained for C_3_.

Yield: 99%. Solid white. Mp: 257 °C. FT-IR-ATR (ZnSe, ν, cm^−1^): 3493, 3340, 3308, 3282 (N-H); 3108, 3060 (C-H, arom.); 1651 (C=N); 1561, 1534, 1513 (C=C); 1191 (C-N); 837 (*p*-subst); 621 (Py-ring). ^1^H NMR (D_2_O, δ, ppm): 8.04 (d, *J* = 7.7 Hz, 4H, **4**); 6.89 (d, *J* = 7.6 Hz, 4H, **3**); 4.33 (t, *J* = 7.2 Hz, 4H, **5**); 2.55 (p, *J* = 7.1 Hz, 4H, **6**). ^13^C NMR (D_2_O, δ, ppm): 158.82 (**2**); 142.41 (**4**); 110.15 (**3**); 54.99 (**5**); 30.50 (**6**). (Spectra in Appendix A).

**1,1′-(Butane-1,4-diyl)bis(4-aminopyridin-1-ium) bromide [C_4_(4-NH_2_Py)_2_][Br]_2_ (C_4_)** [44,45]



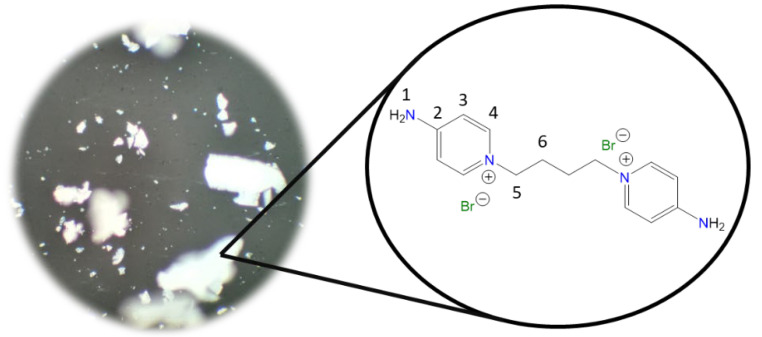



Chemical structure (right) and the optical micrograph (left) of the compound obtained for C_4_.

Yield: 99%. Solid white. Mp: 246 °C. FT-IR-ATR (ZnSe, ν, cm^−1^): 3429, 3366, 3297 (N-H); 3160, 3055 (C-H, arom.); 1666 (C=N); 1561, 1544, 1513 (C=C); 1196 (C-N); 827 (*p*-subst); 669 (Py-ring). ^1^H NMR (D_2_O, δ, ppm): 8.00 (d, *J* = 7.3 Hz, 4H, **4**); 6.88 (d, *J* = 7.5 Hz, 4H, **3**); 4.20 (s, 4H, **5**); 1.91 (s, 4H, **6**). ^13^C NMR (D_2_O, δ, ppm): 158.76 (**2**); 142.46 (**4**); 109.94 (**3**); 57.30 (**5**); 26.73 (**6**). (Spectra in Appendix A).

**1,1′-(Pentane-1,5-diyl)bis(4-aminopyridin-1-ium) bromide [C_5_(4-NH_2_Py)_2_][Br]_2_ (C_5_)** [44]



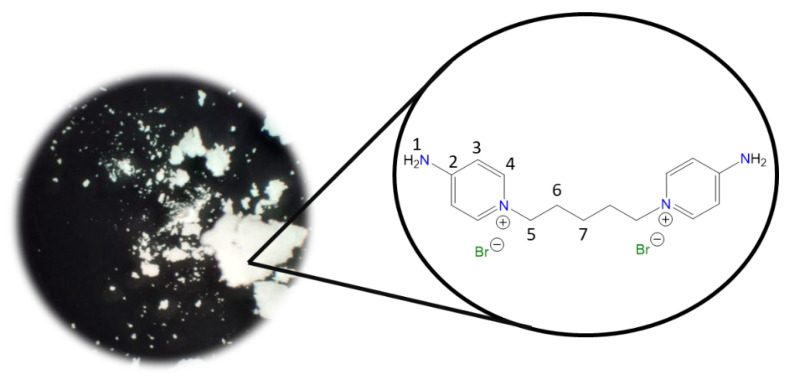



Chemical structure (right) and the optical micrograph (left) of the compound obtained for C_5_.

Yield: 99%. Solid white. Mp: Not observed until heating. FT-IR-ATR (ZnSe, ν, cm^−1^): 3398, 3297 (N-H); 3144, 3060 (C-H, arom.); 2933, 2865 (C-H, aliph.); 1661 (C=N); 1566, 1545, 1513 (C=C); 1192 (C-N); 838 (*p*-subst); 610 (Py-ring). ^1^H NMR (D_2_O, δ, ppm): 8.01 (d, *J* = 7.6 Hz, 4H, **4**); 6.88 (d, *J* = 7.6 Hz, 4H, **3**); 4.17 (t, *J* = 7.0 Hz, 4H, **5**); 1.93 (p, *J* = 7.2 Hz, 4H, **6**); 1.29 (m, 2H, **7**). ^13^C NMR (D_2_O, δ, ppm): 158.61 (**2**); 142.55 (**4**); 109.83 (**3**); 57.63 (**5**); 29.27 (**6**); 21.60 (**7**). (Spectra in Appendix A).

**1,1′-(Hexane-1,6-diyl)bis(4-aminopyridin-1-ium) bromide [C_6_(4-NH_2_Py)_2_][Br]_2_** (C_6_) [44,45]



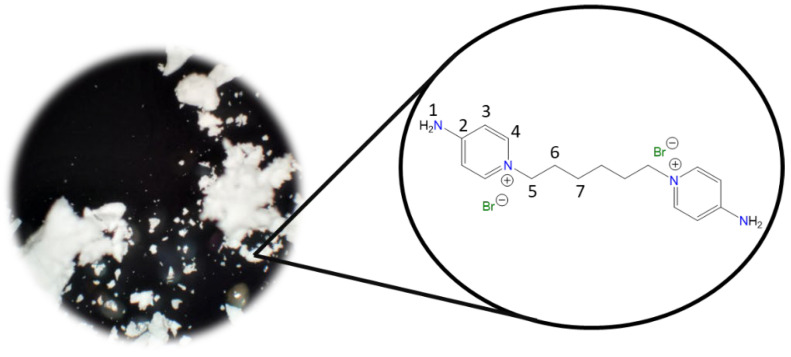



Chemical structure (right) and the optical micrograph (left) of the compound obtained for C_6_.

Yield: 99%. Solid white. Mp: 309 °C. FT-IR-ATR (ZnSe, ν, cm^−1^): 3418, 3308, 3260 (N-H); 3192, 3102, 3049 (C-H, arom.); 2955, 2939, 2860 (C-H, aliph.); 1650 (C=N); 1616, 1556, 1535 (C=C); 1178 (C-N); 845 (*p*-subst); 661 (Py-ring). ^1^H NMR (D_2_O, δ, ppm): 8.03 (d, *J* = 7.6 Hz, 4H, **4**); 6.89 (d, *J* = 7.6 Hz, 4H, **3**); 4.16 (t, *J* = 7.1 Hz, 4H, **5**); 1.86 (p, *J* = 7.2 Hz, 4H, **6**); 1.35 (m, 2H, **7**). ^13^C NMR (D_2_O, δ, ppm): 158.57 (**2**); 142.55 (**4**); 109.84 (**3**); 58.07 (**5**); 29.76 (**6**); 24.93 (**7**). (Spectra in Appendix A).

**1,1′-(Octane-1,8-diyl)bis(4-aminopyridin-1-ium) bromide [C_8_(4-NH_2_Py)_2_][Br]_2_ (C_8_)** [45]



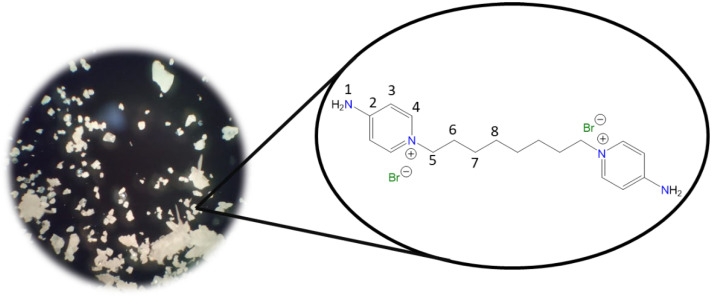



Chemical structure (right) and the optical micrograph (left) of the compound obtained for C_8_.

Yield: 91%. Solid white. Mp: 301 °C. FT-IR-ATR (ZnSe, ν, cm^−1^): 3265, 3223 (N-H); 3192, 3096 (C-H, arom.); 2944, 2913, 2849 (C-H, aliph.); 1655 (C=N); 1561, 1545, 1508 (C=C); 1186 (C-N); 830 (*p*-subst); 648 (Py-ring). ^1^H NMR (DMSO-*d_6_*, δ, ppm): 8.22 (d, *J* = 7.5 Hz, 4H, **4**); 8.14 (s, 4H, **1**); 6.86 (d, *J* = 7.5 Hz, 4H, **3**); 4.10 (t, *J* = 7.2 Hz, 4H, **5**); 1.71 (p, *J* = 6.3 Hz, 4H, **6**); 1.22 (m, 8H, **7**, **8**). ^13^C NMR (DMSO-*d_6_*, δ, ppm): 158.57 (**2**); 142.89 (**4**); 109.36 (**3**); 56.95 (**5**); 30.20 (**6**); 28.26 (**7**); 25.29 (**8**). (Spectra in Appendix A).

**1,1′-(Decane-1,10-diyl)bis(4-aminopyridin-1-ium) bromide [C_10_(4-NH_2_Py)_2_][Br]_2_ (C_10_)** [46,47]



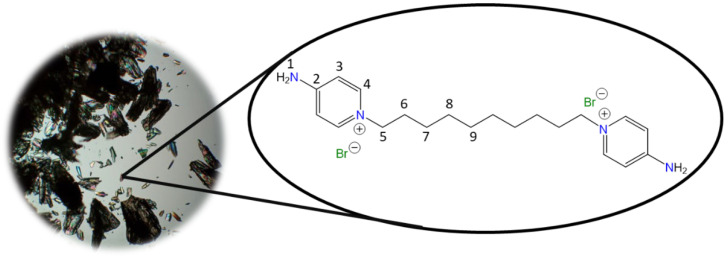



Chemical structure (right) and the optical micrograph (left) of the compound obtained for C_10_.

Yield: 83%. Solid pale orange. Mp: 237 °C. FT-IR-ATR (ZnSe, ν, cm^−1^): 3302, 3229 (N-H); 3113, 3044 (C-H, arom.); 2927, 2854 (C-H, aliph.); 1640 (C=N); 1566, 1534, 1502 (C=C); 1181 (C-N); 848 (p-subst); 610 (Py-ring). ^1^H NMR (DMSO-d_6_, δ, ppm): 8.24 (d, J = 7.5 Hz, 4H, **4**); 8.17 (s, 4H, **1**); 6.88 (d, J = 7.5 Hz, 4H, **3**); 4.12 (t, J = 7.2 Hz, 4H, **5**); 1.70 (p, J = 7.2 Hz, 4H, **6**); 1.19 (m, 12H, **7**, **9**). ^13^C NMR (DMSO-d_6_, δ, ppm): 158.54 (**2**); 142.86 (**4**); 109.30 (**3**); 56.92 (**5**); 30.23 (**6**); 28.68 (**7**); 28.36 (**8**); 25.35 (**9**). (Spectra in Appendix A).

**1,1′-(Dodecane-1,12-diyl)bis(4-aminopyridin-1-ium) bromide [C_12_(4-NH_2_Py)_2_][Br]_2_ (C_12_)** [45,46]



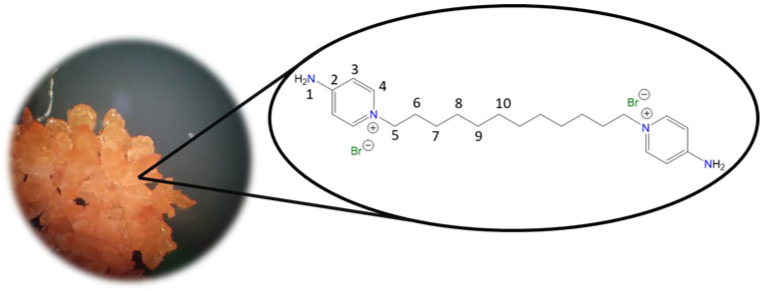



Chemical structure (right) and the optical micrograph (left) of the compound obtained for C_12_.

Yield: 66%. Solid orange. Mp: 210 °C. FT-IR-ATR (ZnSe, ν, cm^−1^): 3312, 3281 (N-H); 3106, 3050 (C-H, arom.); 2922, 2856 (C-H, aliph.); 1643 (C=N); 1567, 1541, 1506 (C=C); 1192 (C-N); 854 (*p*-subst); 618 (Py-ring). ^1^H NMR (DMSO-*d_6_*, δ, ppm): 8.27 (d, *J* = 7.5 Hz, 4H, **4**); 8.19 (s, 4H, **1**); 6.89 (d, *J* = 7.5 Hz, 4H, **3**); 4.12 (t, *J* = 7.2 Hz, 4H, **5**); 1.70 (p, *J* = 7.3 Hz, 4H, **6**); 1.18 (m, 16H, **7**–**10**). ^13^C NMR (DMSO-*d_6_*, δ, ppm): 158.50 (**2**); 142.81 (**4**); 109.22 (**3**); 56.84 (**5**); 30.21 (**6**); 28.77 (**7, 8**); 28.38 (**9**); 25.32 (**10**). (Spectra in Appendix A).


**1,1′-(1,4-Phenylenebis(methylene))bis(4-aminopyridin-1-ium)bromide [Ph(CH_2_)_2_(4-NH_2_ Py)_2_][Br]_2_ (C_Ph_)**




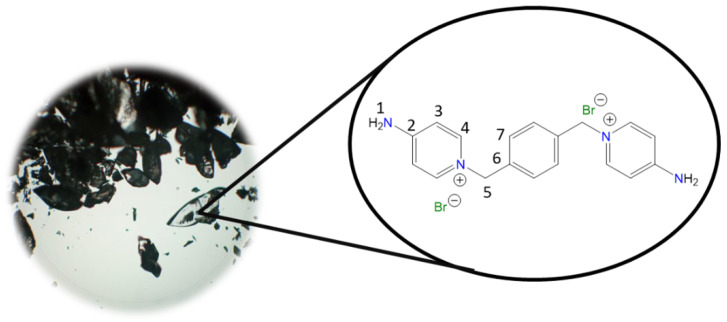



Chemical structure (right) and the optical micrograph (left) of the compound obtained for C_Ph_.

Yield: 99%. Solid white. Mp: 345 °C. FT-IR-ATR (ZnSe, ν, cm^−1^): 3391, 3260 (N-H); 3144, 3043, 3018 (C-H, arom.); 2992, 2886 (C-H, aliph.); 1666 (C=N); 1562, 1544, 1510 (C=C); 1171 (C-N); 877, 857 (*p*-subst); 818, 759 (*p*-subst); 600 (Py-ring). ^1^H NMR (DMSO-*d_6_*, δ, ppm): 8.05 (d, *J* = 7.6 Hz, 4H, **4**); 7.42 (s, 4H, **7**); 6.87 (d, *J* = 7.6 Hz, 4H, **3**); 5.36 (s, 4H, **5**). ^13^C NMR (DMSO-*d_6_*, δ, ppm): 158.93 (**2**); 142.67 (**4**); 135.39 (**6**); 128.89 (**7**); 110.00 (**3**); 60.36 (**5**). (Spectra in Appendix A).

### 3.3. Solubility Test

10 mg of the organic salt was placed in a 5 mL test tube. Then, 0.5 mL of the respective solvent was added and shaken vigorously at room temperature for 1 min. If solid or traces of salt were observed after stirring, the mixture was heated to 40 °C and shaken again vigorously for 1 min. If the salt dissolved completely without applying heat, it was called soluble (+). If the solid completely dissolves at 40 °C and precipitates again at room temperature, it is called soluble in heat. (++). If traces were observed after heating to 40 °C, it was called partially soluble (+/−). If a similar or equivalent amount of solid was observed as initially added after heating to 40 °C, the salt was called insoluble (−).

### 3.4. Bacterial Strains and Culture Media

*S. aureus* strains ATCC 6538, methicillin-resistant *S. aureus* (MRSA) 97-7, *E. coli* ATCC 25922, and *S. enterica serovar Typhimurium* ATCC 14028s were used. All strains were cultured on Luria Bertani (LB) agar and incubated at 37 °C for 18–24 h. All strains were kindly donated by Marcela Wilkens (Universidad de Santiago de Chile, Santiago, Chile).

### 3.5. Antibiotic Susceptibility Assays

The antibiotic susceptibility of all bacterial strains was determined following the disc agar diffusion assay described by Clinical Laboratory Standards (CLSI). The strains were cultured overnight (ON) and subsequently diluted in Mueller–Hinton broth (MHB) to a McFarland of 0.5 (1 × 10^8^ colony-forming unit (CFU)/mL) and seeded homogeneously on Petri dishes containing Mueller–Hinton agar (MHA). Sterile discs containing different concentrations of antibiotics were placed on the inoculated agar. After incubation for 18 h at 37 °C, the inhibition diameters were measured, and these values, in millimeters (mm), were interpreted according to the criteria established by CLSI as resistant (R) or sensitive (S). All experiments were done in triplicate, in three individual experiments. The antibiotics used for susceptibility testing on strains were penicillin (10 OUF), oxacillin (1 μg), ampicillin (10 μg), cephalothin (30 μg), cefuroxime (30 μg), ciprofloxacin (5 μg), clindamycin (2 μg), gentamicin (10 μg), tetracycline (30 μg), and erythromycin (15 μg).

### 3.6. Minimal Inhibitory Concentration Determination

The minimal inhibitory concentration (MIC) was defined as the lowest concentration of a compound for which no growth was observed. The MIC of different organic salts was determined according to a procedure established by CLSI guidelines using the microbroth dilution method in 96-well plates. Briefly, bacterial strains were incubated ON in 3 mL MHB at 37 °C with shaking at 220 rpm. The bacterial cultures were diluted in a phosphate-buffered solution (PBS) to McFarland 0.5 (1 × 10^8^ CFU/mL). To each well, 188 μL of MHB, 10 μL of each compound (0.05 to 2000 μg/mL diluted in distilled water), and finally 2 μL bacterial suspension at McFarland 0.5 (1 × 10^8^ CFU/mL) was added to complete a final volume of 200 μL. In addition, some wells were used as growth and sterility controls. The plates were incubated at 37 °C for 24 h, and the optical density was measured at 600 nm in an ELISA lector Synergy HT Multi-Detection Microplate Reader (BioTek^®^, Winooski, VT, USA). Results are expressed in μg/mL, and all experiments were done in triplicate, in three individual experiments.

### 3.7. Live/Dead Assays

#### Preparation Live/Dead Assays

The LIVE/DEAD BacLightTM Bacterial Viability Kit (L7012) was purchased from Invitrogen (Waltham, MA, USA). The cultures were prepared based on the guidelines of the BacLight Kit [64]. Briefly, the bacteria selected for this study (*E. coli* ATCC 25922 and *S. aureus* MRSA 97-7) were grown in 30 mL of Luria Bertani (LB) broth to late log phase. Subsequently, the bacterial cultures were concentrated by centrifugation at 10,000 rpm for 15 min; then, the supernatant was removed, and the pellet was resuspended in 2 mL of a 0.85% aqueous NaCl solution. Subsequently, the controls for live cells were prepared by adding 1 mL of bacterial suspension to 20 mL of a 0.85% aqueous NaCl solution, and the control for dead cells (1 mL of bacterial suspension in 20 mL of a 70% *v*/*v* aqueous isopropanol solution). Additionally, the treatments were prepared, for which ½ and ¼ of the MIC of the C_10_ compound was used (1 mL of bacterial suspension in 20 mL of a 0.85% aqueous NaCl solution and 1 mL of C_10_). Subsequently, the samples were incubated at 37 °C for 3 h under constant agitation. After incubation, the cultures were centrifuged at 10,000 rpm for 15 min. Then, the pellets were resuspended in 10 mL of a 0.85% aqueous NaCl solution, and the optical density at 670 nm (OD_670_) was determined. For fluorescence measurement, the cultures were adjusted to 2 × 10^8^ bacteria/mL (OD_670_~0.06) for the Gram-negative bacteria used (*E. coli* ATCC 25922) and to 2 × 10^7^ bacteria/mL (OD_670_~0.30) for Gram-positive bacteria (*S. aureus* MRSA 97-7). Then, 100 µL of each sample (live cell controls, dead cell controls, and treatments with ½ MIC and ¼ of C_10_ compound) was placed in a 96-well plate for fluorescence and mixed with 100 µL of Live/Dead 2X. The samples were incubated in the dark for 15 min at room temperature. Finally, fluorescence was measured in a fluorimeter, Synergy HT Multi-Detection Microplate Reader (BioTek^®^), using a 488/20 nm excitation filter (for both SYTO9 and PI) and a 528/20 nm (SYTO9 emission wavelength) and 645/40 nm (PI emission wavelength) for each well. Live and dead cells can be differentiated based on the relative green and red fluorescence. A standard curve for which the SYTO9 (G: green) to PI (R: red) fluorescence ratio (G/R ratio) was used to calculate the percentage of live/dead cells. The determination of the live/dead ratio was performed by dividing the fluorescence intensity obtained at 528 nm (G: green) by that measured at 645 nm (R: red). All experiments were done in triplicate, in three individual experiments.

### 3.8. Cytotoxicity Assays

#### 3.8.1. HeLa Cell Culture

In order to determine the cytotoxic potential of C_10_ and C_12_ salts (compounds that showed the greatest antibacterial activity), the HeLa cervical cancer cell line (ATCC CCL-2, USA American Type Culture) was used. The cells were maintained in 5% CO_2_ at 37 °C in Eagle’s Minimum Essential Medium (MEM, Hyclone™) supplemented with 10% (*v*/*v*) fetal bovine serum (FBS) (Hyclone™), 4 mM L-glutamine (Corning^®^), 1X Penicillin/Streptomycin 100X (Corning^®^), and 2.5 mg/mL Amphotericin B (Corning^®^). Cell propagation was performed using 0.25% Trypsin-EDTA (Corning^®^) at a 1:20 ratio relative to the final volume of the flask to be propagated. The flasks were left for 5 min at 37 °C to facilitate detachment of the cell monolayer, and trypsin was inactivated with 9.5 mL of supplemented MEM medium for each 0.5 mL of trypsin used. Subculturing of the cells was carried out at a 1:10 ratio relative to the final flask volume. Cells were observed daily under an AE 2000 inverted microscope (Motic, Universal City, TX, USA).

#### 3.8.2. Cytotoxicity Evaluation

The cytotoxicity evaluation of the salts C_10_ and C_12_ at concentrations of 62.5, 31.2, 15.6, and 7.8 μg/mL, and of the C_12_ vehicle (0.3% (*v*/*v*) DMSO), was performed by quantifying lactate dehydrogenase (LDH) in the medium.

The highest concentrations tested correspond to the MIC value. 31.2 μg/mL in *S. aureus* ATCC 6538, *S. aureus* MRSA 97-7, *S*. *Typhimurium* ATCC 14028s, and a MIC of 62.5 μg/mL in *E. coli* ATCC 25922 for both salts studied. The concentrations of 15.6 and 7.8 μg/mL correspond to ½ and ^1^/_4_ of the MIC of the strains studied. Compound C_10_ was diluted in the same MEM medium supplemented with 5% FBS; therefore, its vehicle was not evaluated. Briefly, HeLa cells were incubated in flat-bottom 96-well plates at a concentration of 1600 cells/well in MEM medium supplemented with 5% FBS. The cells were allowed to adhere overnight (ON), washed twice with phosphate-buffered saline (PBS), and then incubated with 100 μL of the compounds. Death and live controls were incubated with MEM medium supplemented with 5% FBS. Cytotoxicity was evaluated at 6, 12, and 24 h post-incubation with the solutions. Free LDH measurement was performed using the Pierce LDH Cytotoxicity Assay Kit (Thermo Scientific, Waltham, MA, USA). For death controls, 10 μL of lysis buffer was added and incubated for 30 min at 37 °C. Then, 100 μL of working solution was added to all wells and incubated for 30 min at room temperature in the dark. The reaction was stopped with 50 μL of stop solution, and colorimetric changes were measured at 490 nm using a spectrophotometer (Nanoquant Infinite M200 pro, TECAN™, AG, Männedorf, Switzerland).

## 4. Conclusions

This work is part of the development of molecules aimed at addressing the growing increase in bacterial resistance. The evaluated molecules, bis(4-aminopyridinium) salts, with systematic variations in the length of their aliphatic chain, allowed for the identification of structural patterns that favor this activity. Analysis of the physicochemical properties of the prepared salts revealed that these molecules possess high thermal stability and high solubility in polar solvents such as DMSO, DMF, MeOH, EtOH, and especially in water, favorable conditions for their application in biological environments and their use as disinfectants.

Among the salts evaluated, C_10_ and C_12_ showed the highest antibacterial activity against *S. aureus* strains (including MRSA), *E. coli*, and *Salmonella enterica* serovar Typhimurium, even at concentrations below 65 μg/mL. Furthermore, these molecules exhibited low cytotoxicity in human cells, favoring their application in the formulation of new disinfectants for use on hospital surfaces. The correlation between alkyl chain length and antibacterial activity reinforces what has been described in the literature: a longer aliphatic chain increases hydrophobic interaction with the bacterial membrane, favoring its rupture. The live/dead assay confirmed that one of the sites of action of the studied compounds is the bacterial cytoplasmic membrane, which would increase its permeability in both Gram-positive and Gram-negative bacteria.

The results obtained open the possibility of advancing the evaluation of a promising candidate for the development of a new disinfectant agent for hospital use, C_10_, considering the bacterial models used in this study and the analyses performed. However, analyses are still needed to definitively determine its mechanism of action, activity against biofilms, effects on hospital surfaces, and other chemical–physical properties necessary for its use.

## Figures and Tables

**Figure 1 molecules-30-03962-f001:**
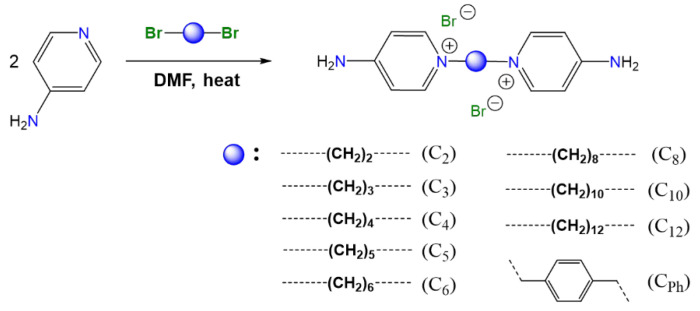
Synthesis of organic salts.

**Figure 2 molecules-30-03962-f002:**
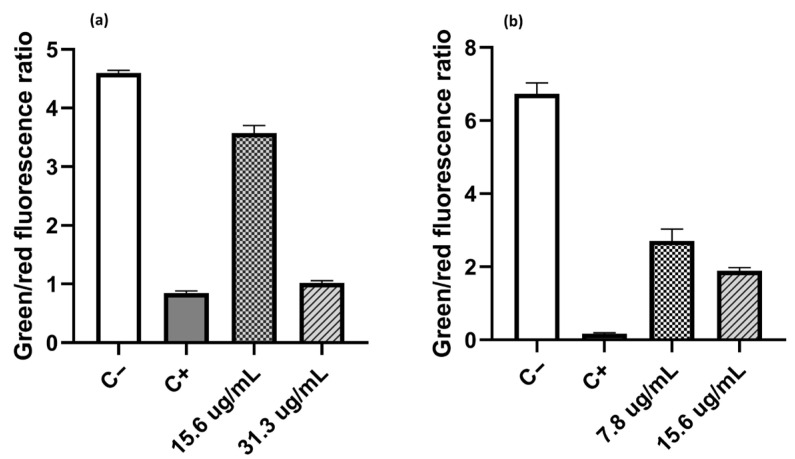
Effect of C_10_ on membrane cytoplasmic in (**a**) *E. coli* ATCC 25952 and (**b**) *S. aureus* MRSA 97-7. Isopropanol 70% *v*/*v* was used as a positive control (C+; 100% dead cells), and an untreated bacterial culture was used as a negative control (C−; 100% live cells). Treatments were carried out with C10 at concentrations of ½ and ¼ of the MIC for each of the bacterial models.

**Figure 3 molecules-30-03962-f003:**
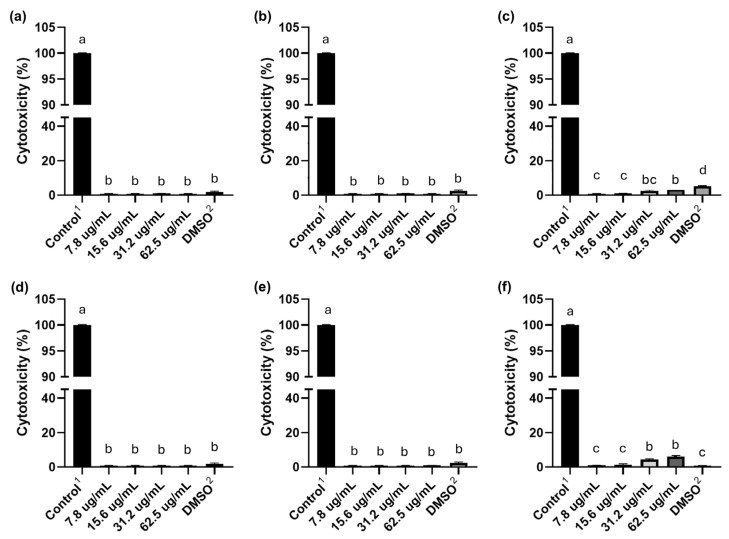
Cytotoxicity of HeLa cells exposed to different concentrations of C10 and C12 salts. Cytotoxicity was determined after 6, 12, and 24 h of incubation of the cells. The LDH assay as an indicator of cell death was used. The effect of C10 salt on HeLa cells at different incubation times is shown in Figures (**a**), (**b**), and (**c**) for 6, 12, and 24 h, respectively. (**d**), (**e**), and (**f**) represent the effect of C12 salt after 6, 12, and 24 h of incubation, respectively. The concentrations tested correspond to the MIC, ½, and ¼ of the MIC of all the strains studied. 1 Death control, 10 μL of lysis buffer incubated for 30 min at 37 °C. 2 DMSO (0.3% (*v*/*v*) was used as a solvent control for C12 salt. Shapiro–Wilk test indicated non-normal distribution (*p* < 0.05 for all panels). Therefore, the Kruskal–Walli test was applied, showing significant differences among treatments (*p* < 0.001). Post hoc pairwise comparisons were performed using the Mann–Whitney U test with Bonferroni correction. Different letters (a, b, c, d) indicate statistically significant differences between treatments.

**Table 1 molecules-30-03962-t001:** Summary of solubility tests.

	Solvent									
Salt ^a^	H_2_O	MeOH	EtOH	DMSO	DMF	AcCN	Acetone	THF	CHCl_3_	Et_2_O
**C_2_**	+	−	−	−	−	−	−	−	−	−
**C_3_**	+	+	+/−	+	++	−	−	−	−	−
**C_4_**	+	+	−	+	−	−	−	−	−	−
**C_5_**	+	+	++	+	+/−	−	−	−	−	−
**C_6_**	+	+	−	+	−	−	−	−	−	−
**C_8_**	+	+	+/−	+	−	−	−	−	−	−
**C_10_**	+	+	−	+	++	−	−	−	−	−
**C_12_**	++	+	+	+	++	−	−	−	−	−
**C_Ph_**	+/−	−	−	+/−	−	−	−	−	−	−

^a^ Concentration: 20 mg/mL; +: soluble at room temperature (20–25 °C); −: insoluble at 40 °C; +/−: partially soluble at 40 °C; ++: soluble at 40 °C.

**Table 2 molecules-30-03962-t002:** Summary of thermal properties.

Salt	Mp (°C) ^a^	T_onset_ (°C) ^b^	T_5%_ (°C) ^c^	T_10%_ (°C) ^d^	T_50%_ (°C) ^e^	T_d1_ (°C) ^f^	T_d2_ (°C) ^f^	R (%) ^g^
**C_2_**	397	389	400	404	416	410	525	20.0
**C_3_**	257	324	331	341	370	368	533	2.0
**C_4_**	246	333	336	344	372	374	-	1.7
**C_5_**	-	330	336	344	371	371	534	1.0
**C_6_**	309	344	344	350	373	372	500	1.7
**C_8_**	301	345	345	353	377	378	521	1.7
**C_10_**	237	329	335	343	368	368	457	2.6
**C_12_**	210	318	336	345	369	374	539	0.0
**C_Ph_**	345	307	332	338	358	358	504	0.0

^a^ Melting point. ^b^ Temperature at which the polymer degradation begins. ^c, d, e^ Temperature at which 5%, 10% and 50% weight is lost, respectively. ^f^ Temperature at which the degradation rate is maximum. ^g^ Percentage of residual material.

**Table 3 molecules-30-03962-t003:** Study of the susceptibility to antibiotics of different classes of strains.

Antibiotic	Strains			
	*S. aureus* ATCC 6538	*S. aureus* MRSA 97-7	*E. coli* ATCC 25922	*S. Typhimurium* ATCC 14028s
Penicillin	R	R	R	R
Oxacillin/Ampicillin	S	R	R	R
Cephalothin	S	R	R	R
Cefuroxime	S	R	R	R
Ampicillin/Sulbactam	S	R	S	S
Ciprofloxacin	S	R	S	S
Clindamycin	S	R	R	S
Gentamicin	S	R	S	S
Tetracycline	S	R	R	R
Erythromycin	S	R	R	R

S: The bacterial strain was susceptible to the antibiotics tested. R: The bacterial strain was resistant to the antibiotics tested.

**Table 4 molecules-30-03962-t004:** Evaluation of the antibacterial activity of the organic salts against *S. aureus* strains, *E. coli,* and *S. Typhimurium* strains.

Strains	MIC (μg/mL)								
	C_Ph_	C_2_	C_3_	C_4_	C_5_	C_6_	C_8_	C_10_	C_12_
*S. aureus* ATCC 6538	1000	1000	>2000	250	>2000	500	250	31.2	31.2
*S. aureus* MRSA 97-7	>2000	>2000	>2000	250	>2000	2000	1000	31.2	31.2
*E. coli* ATCC 25922	>2000	>2000	>2000	250	2000	250	250	62.5	62.5
*S. Typhimurium* ATCC 4028s	>2000	>2000	>2000	250	>2000	500	500	31.2	31.2

## Data Availability

The original contributions presented in this study are included in the article/Appendix A. Further inquiries can be directed to the corresponding author.

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
