# Peer review of "Antibacterial Activity of Bis(4-aminopyridinium) Compounds for Their Potential Use as Disinfectants"

_molecules, 2025, doi:10.3390/molecules30193962_

Round 1
Reviewer 1 Report
Comments and Suggestions for Authors
The authors prepared a series of organic salts based on 4-aminopyridinium and characterized them using NMR, FT-IR, thermogravimetric analysis (TGA), differential scanning calorimetry (DSC), and solubility tests. These salts were subsequently shown to exhibit enhanced antibacterial activity with increasing alkyl chain length.
I would like to offer the following suggestions:
1) The reported reaction yield appears to be quite high. I am curious about the reactivity of the free amine group—what is the main byproduct of this reaction?
2) Based on my experience, biphenyl often serves as a good alternative to long alkyl chains. I would recommend testing biphenyl instead of phenyl in this context.
3) The English writing should be polished throughout the manuscript to improve clarity and readability.
Comments on the Quality of English LanguageI understand that the author may not be a native English speaker; however, the manuscript would benefit from thorough language editing to improve clarity and readability throughout.
Author Response
Please refer to the attached PDF file for the response to the reviewer.

Reviewer 2 Report
Comments and Suggestions for Authors
This study synthesized an organic salt using conventional methods, presenting a potential alternative to currently employed hospital disinfectants, though both experimental design and manuscript quality require improvement. Specific comments are as follows:
1. Lines 20-22: The statement "Antibiotic susceptibility testing confirmed that the different strains studied are resistant to different classes of antibiotics (MDR)..." is overly generalized, given the well-documented diversity of bacterial resistance mechanisms and antibiotic target sites.
2. Lines 52-57: The authors failed to adequately describe the inherent limitations of current hospital disinfectants, which consequently obscures the innovative potential of the synthesized compound.
3. Lines 114-120: The results section lacks appropriate reference to supporting figures (e.g., Fig. S1). Similar omissions should be systematically addressed throughout the manuscript.
4. Lines 160-162: While excellent aqueous solubility represents a hallmark characteristic of effective solid disinfectants, the poor solubility of the synthesized compound raises concerns regarding its practical viability as a conventional disinfectant replacement.
5. Lines 241-244: The proposed correlation between C4-mediated bactericidal activity and cell membrane damage remains unclear. The original charge distribution of C4 should be explicitly characterized to substantiate this mechanism.
6. Lines 307-310: The reliance solely on live/dead staining cannot definitively establish a single antimicrobial target. Membrane damage may represent a secondary consequence of drug-induced oxidative stress (either endogenous or exogenous). The experimental design appears to lack mechanistic validation, relying excessively on speculative interpretation of endpoint results. The extensive analogy to antibiotic mechanisms warrants complementary investigation of the organic salt's specific antimicrobial targets, as demonstrated in Yang's (2025) (DOI: 10.1016/j.ijfoodmicro.2025.111071) work.
7. Lines 369-372: Given that quaternary ammonium salts are known to induce bacterial resistance, similar concerns may apply to the synthesized compound. A comparative analysis of advantages/disadvantages relative to current clinical disinfectants should be included.
8. Line 502: Was DMSO employed during compound solubilization?
Author Response

(The authors gave the same response as above.)

Reviewer 3 Report
Comments and Suggestions for Authors
The authors described the synthesis of nine bis(4-aminopyridinium) salts with different alkyl chain lengths and their spectroscopic characterization. However, it should be made clear that these compounds are not new chemical structures, but known substances that the authors have used in the context of biological research. The current form of the manuscript may be misleading, suggesting that new chemical compounds have been described. The synthesis and characterization data should be limited in the body of the manuscript and moved to supplementary material (or omitted altogether with a note that compounds were synthesized according to known literature procedures), while it should be made clear in the text that known compounds were synthesized to evaluate their biological properties.
The novelty of the work is the use of these salts as potential disinfectants, especially against MDR strains. The results indicate moderate antimicrobial activity (MICs higher than those of classical disinfectants), but within the range useful for surface disinfection formulations. Advantages include good solubility in water (especially salts with shorter alkyl chains) and high thermal stability.
No in-house cytotoxicity studies are available. The authors rely only on literature data. Before publication, especially in a journal with a high IF, it is recommended to carry out at least basic cytotoxicity tests, e.g. on skin or lung cells, to better assess the safety of these compounds.
There are no data on chemical stability in aqueous solutions. For disinfection applications, the stability of compounds in the presence of organic contaminants and under different pH conditions is crucial. It is recommended that appropriate stability tests be performed under conditions that simulate practical applications.
Activity tests against biofilms are also worth considering.
Author Response

(The authors gave the same response as above.)

Round 2
Reviewer 3 Report
Comments and Suggestions for Authors
The authors have put a great deal of work into revising and improving the manuscript, both in terms of the experimental section and the content. In my opinion, it is suitable for publication in its current form.